# Expressed fusion gene landscape and its impact in multiple myeloma

A. Cleynen[1], R. Szalat[2], M. Kemal Samur[2,3], S. Robiou du Pont[4], L. Buisson[4], E. Boyle[4], M.L. Chretien[5], K. Anderson[2], S. Minvielle [6], P. Moreau[6], M. Attal[4], G. Parmigiani[2,3], J. Corre[4], N. Munshi[2,7] & H. Avet-Loiseau[4]

Multiple myeloma is a plasma cell malignancy characterized by recurrent *IgH* translocations and well described genomic heterogeneity. Although transcriptome profiles in multiple myeloma has been described, landscape of expressed fusion genes and their clinical impact remains unknown. To provide a comprehensive and detailed fusion gene cartography and suggest new mechanisms of tumorigenesis in multiple myeloma, we performed RNA sequencing in a cohort of 255 newly diagnosed and homogeneously treated multiple myeloma patients with long follow-up. Here, we report that patients have on average 5.5 expressed fusion genes. Kappa and lambda light chains and *IgH* genes are main partners in a third of all fusion genes. We also identify recurrent fusion genes that significantly impact both progression-free and overall survival and may act as drivers of the disease. Lastly, we find a correlation between the number of fusions, the age of patients and the clinical outcome, strongly suggesting that genomic instability drives prognosis of the disease.

[1] Institut Montpellierain Alexander Grothendieck, CNRS, Univ. Montpellier, Montpellier 34090, France. [2] Medical Oncology, Dana-Farber Cancer Institute, Harvard Medical School, Boston, MA 02115, USA. [3] Department of Biostatistics and Computational Biology, Dana-Farber Cancer, Institute, Harvard Medical School, Boston, MA 02115, USA. [4] IUC-Oncopole, and CRCT INSERM U1037, Toulouse 31100, France. [5] CHU Dijon, Dijon 21000, France. [6] CHU Nantes, Nantes 44000, France. [7] VA Boston Healthcare System, West Roxbury, MA 02215, USA. Correspondence and requests for materials should be addressed to N.M. (email: nikhil_munshi@dfci.harvard.edu) or to H.A.-L. (email: avet-loiseau.h@chu-toulouse.fr)

**M**ultiple myeloma (MM) is a plasma cell malignancy associated with production of monoclonal immunoglobulin and a variable clinical presentation with symptoms related with anemia, hypercalcemia, bone lesions, and renal involvement[1]. Myeloma is characterized by heterogeneity at multiple levels with a complex subclonal structure and driver mutations and a diverse mutational spectrum across patients with distinct patterns of clonal evolution[2, 3]. A large-scale expression profile has identified major signaling pathways operative in MM and some of the associated genes have been identified as important mediators of myelomagenesis. These transcriptome analyses have identified 10 subgroups with different clinical and biological behaviors[4–6].

Based on cytogenetics and fluorescent in situ hybridization (FISH) analysis, MM is divided into hyperdiploid MM (HMM) and non-hyperdiploid MM (NHMM) with majority of recurrent translocations usually occurring at the immunoglobulin *IGH* gene locus (14q32)[7]. Such translocations, observed in about 50% of patients, mostly NHMM patients, frequently involve t(11;14) (~20% of patients), t(4;14) (~15–20% of patients), t(8;14), t(14;16), and t(14;20) (~2% of patients each)[8]. These translocations usually drive over-expression of the partner genes, *CCND1*, *FGFR3*, *MYC*, *c-MAF*, and *MAFB*, respectively. HMM tends to have a better prognosis than NHMM or *IgH* translocated MM, which have a more heterogeneous evolution[8]. However, limited data is available so far regarding non *IgH* translocations, with the exception of those involving the Kappa and Lambda or *MYC* loci[9, 10]. In particular, it is unclear whether non-recurrent translocation events are common events in HMM. Moreover, only the t(4;14) generates a fusion protein whose length depends on the breakpoints at the DNA level. The *MMSET* and *FGFR3* genes involved in this translocation and in the fusion product play a significant role in disease behavior including their transforming ability and impact on MM cell growth and survival as well as clinical outcome[11–14]. However, impact of other overt or cryptic translocations on generation of fusion products remains unclear.

Over the last decade, next-generation sequencing technologies, including RNA sequencing (RNA-Seq), has allowed for comprehensive investigation of the transcriptome, quantifying gene expression or identifying novel fusion transcripts[15]. Here, we have performed RNA-Seq on a large cohort of newly diagnosed MM patients (with complete clinical and FISH data), and on 71 MM cell lines to study the global fusion landscape in this disease. We report that fusion transcripts are frequent in MM and oftentimes involve kappa or lambda light chain as the main partners of fusions and have significant biological and clinical impact.

## Results

In the IFM-DFCI (Intergroupe Francophone du Myélome - Dana Farber Cancer Institute) cohort of 255 patients, MapSplice reported 13,134 while TopHat reported 48,463 fusion events; in the 71 sequenced cell lines, MapSplice reported 6994 and TopHat 6559 fusion events. For a conservative assessment, we focused on events reported by both algorithms. We performed further processing, including removal of same gene-family fusion or computation of fusion partners' gene homology, to limit the number of false positives (see "Methods" section). This resulted in a list of 1408 candidate fusions in the patient cohort, and 436 in the cell lines. Supplementary Figure 1 illustrates the algorithms' output and their intersection before and after filtering, while Supplementary Data 1 gives the list of all reported fusions including a measurement of the homology of the sequences that surround the breakpoint.

**Distribution of fusions across individuals.** While all cell lines expressed at least one fusion candidate, only 225 patients (88.3%) allowed identification of one (or more) fusion transcript

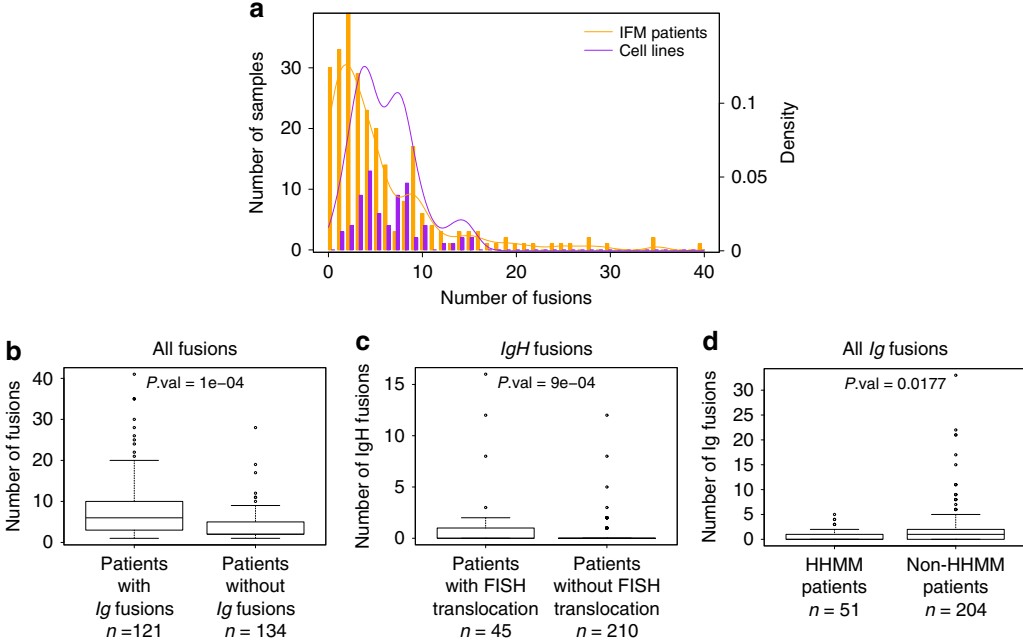

**Fig. 1 Distribution of fusions across individuals. a** Boxplot of the distribution of the number of fusions per patient across samples. Cell lines have at least one fusion and a larger amount of fusions on average. In total, 11.8% of patients have no fusions. The distribution in patients has a heavier right-tail. **b** Number of fusions among patients who have at least one fusion. Patients who have an *Ig* fusion have significantly more fusions than others, conditional on their having at least one fusion (average = 8.3 vs. 3.8). **c** Number of *IgH* fusions. Patients who have a known translocation (t(11;14), t(4;14), and t(14;16)) have significantly more *IgH* fusions than others (average = 1.3 vs. 0.3). **d** Number of *Ig* fusions. High-HMM patients (at least 53 chromosomes) have significantly fewer *Ig* fusions than others (average = 0.65 vs. 2). All *P*-values were computed using permutation tests

expression. Figure 1a shows the distribution of the number of fusions across individuals, indicating that while the average number of fusion per sample is roughly the same (5.5 in patients, 6.1 in cell lines), the distribution is more skewed in cell lines. In particular, 19 patients (7% of the cohort) have more than 15 fusions, which is the maximum number observed in cell lines.

Among patients with at least one fusion, those who have an *Ig* fusion (involving any *IgH* segment, or kappa or lambda light chains) have significantly higher number of other fusions than those without *Ig* fusions (Fig. 1b). We identified different fusion patterns in the previously described MM subgroups, in line with the different clinical and biological behaviors reported. For instance, patients having a known FISH detected translocation t (11;14), t(4;14), and t(14;16), have significantly higher number of *IgH* fusions than those without FISH reported translocations (Fig. 1c), and patients with high HMM (defined as patients having more than 53 chromosomes) have significantly less *Ig* fusions than others (Fig. 1d). Surprisingly, HMM patients do not have significantly less overall fusions than others. Comparing the number of fusions per cytogenic group of patients (Ig-translocated, high-hyperdiploid, deletion 13, hypodiploid and other patients), showed that Ig-translocated and deletion 13 patients tend to have more fusions than others (Supplementary Fig. 2a), while the number of fusions increases with age (Supplementary Fig. 2b).

**Landscape of fusion genes in MM.** Out of the 1408 fusions observed in MM patients, only 559 consist of different pairs of partner genes (we will refer to those as unique fusions), with some candidates observed in as many as 115 patients. Even though almost half of the patients present an *Ig* fusion, these only represent 36% of unique fusions (Tables 1 and 2). This contrast is greater in cell lines with almost 60% expressing an *Ig* fusion while they represent less than a fifth of the total fusions. Overall, 29% of unique fusions are found in at least two patients and are referred to as recurrent, and ~17% fusions are recurrent but do not involve the immunoglobulin genes. Importantly, in both sets, more than half of the remaining (non-recurrent) fusions involve genes located on the same chromosome and separated by less than 10 million base pairs suggesting that the possible mechanisms behind the fusions are associated with in situ chromosomal rearrangements (amplifications, inversions, deletions, tandem duplications, etc.) or read-through fusions.

**IGH–MMSET fusion.** t(4;14) translocation is one of the recurrent translocations involving genes with well-described biological as well prognostic impact in myeloma and is also one of the most frequent expressed fusion genes. In our cohort, 21 patients out of 255 were identified by FISH as presenting a t(4;14) translocation. Of these we could identify 20 by both algorithms. Interestingly, the patient that could not be identified with either algorithms was characterized by a low expression of *MMSET* compared to other patients with this fusion (normalized log-expression of 5.37 compared to an average of 10.53 with standard deviation 1.50). We identified five additional patients with *IGH–MMSET* fusions using our sequencing analysis that had not been detected by FISH, and another three by one algorithm only (one by Mapslice, two by TopHat). Of note, most of these *IGH–MMSET* fusions were identified by the algorithms based on relatively smaller number of reads (dozens compared to hundreds or sometimes thousands). Figure 2 displays the expression of *MMSET* and the supporting number of reads (on a log scale) for patients identified as t(4;14) by either FISH or at least one algorithm.

We performed real-time (RT)-polymerase chain reaction (PCR) to confirm the presence of t(4;14) translocation on a

**Table 1** Distribution of fusion types in patients and cell lines. Recurrent fusions are defined as those occurring in at least two samples in the same population, and percentages are given in terms of percentages of different pairs of partner genes (unique fusions). More than 80% of fusions are specific to one cell line, but only 64% of fusions found in patients are patient-specific. Among unique fusions, more than half are same-chromosomes and less than 10 million bases apart, suggesting read-through fusions, deletions or in situ amplifications

| | IFM | Cell lines |
|---|---|---|
| *Ig* fusions | 36% | 17.7% |
| Recurrent fusions | 28.8% | 11.6% |
| Recurrent non-*Ig* fusions | 17.2% | 7.1% |
| Others | 46.9% | 75.2% |
| Other—same chr | 54.2% | 66.6% |

**Table 2** Percentage of individuals with identified *Ig* fusion transcripts in different groups. Almost half of the cohort have at least one *Ig* fusion, almost 60% in cell lines. There are a large number of fusions involving Kappa (79 patients, all but one have kappa phenotype). Fusions involving Lambda are surprisingly rare, even in the Lambda phenotype patients

| | All IFM | *IgK* patients | *IgL* patients | Cell lines |
|---|---|---|---|---|
| *Ig* fusions | 47.3% | 49% | 42.6% | 59.1% |
| *IgH* fusions | 22.7% | 18.2% | 29.4% | 40.8% |
| *IgK* fusions | 31% | 47.8% | 1.5% | 21.1% |
| *IgL* fusions | 4.3% | 0% | 14.5% | 8.4% |

selection of 26 patients and compared to FISH and both the algorithms (bottom part of Fig. 2 and Supplementary Fig. 3a). In particular, the FISH positive patient who was not detected by either algorithm could not be confirmed by PCR possibly suggesting a false positive, and only one patient identified by both algorithms could not be validated by PCR, possibly due to lack of starting material. Moreover, when PCR and RNA-Seq data were both positive (24 patients), they showed perfect agreement in terms of classification into subtypes of t(4;14) that relate to the three different breakpoints (MB4-1, MB4-2, and MB4-3); confirming the possibility to precisely identify the intron of breakpoint localization with sequencing technologies.

In addition, we confirmed the presence of an *IGH–MMSET* fusion in 19 out of the 20 MM cell lines (including two independent versions of KMS11) that had previously been shown to be t(4;14) by PCR or FISH experiments. KMS34, which was missed by our analysis, was identified by TopHat only with evidence of the fusion occurring within the fourth *IGHM* intron as previously reported (see Venn diagram of Supplementary Fig. 4).

**Landscape of *Ig* partner genes.** Surprisingly, we identified a greater number of fusions involving the kappa light chain compared to *IgH*. Seventy-nine patients (31%) had at least one *IGK* fusion, out of which 76 were known to express a kappa phenotype (48% of all known kappa phenotypes), two were unknown for light-chain phenotype, and only one was known to express a lambda phenotype. We found a similar pattern for lambda fusions (Table 2). Novel recurrent partners were identified as

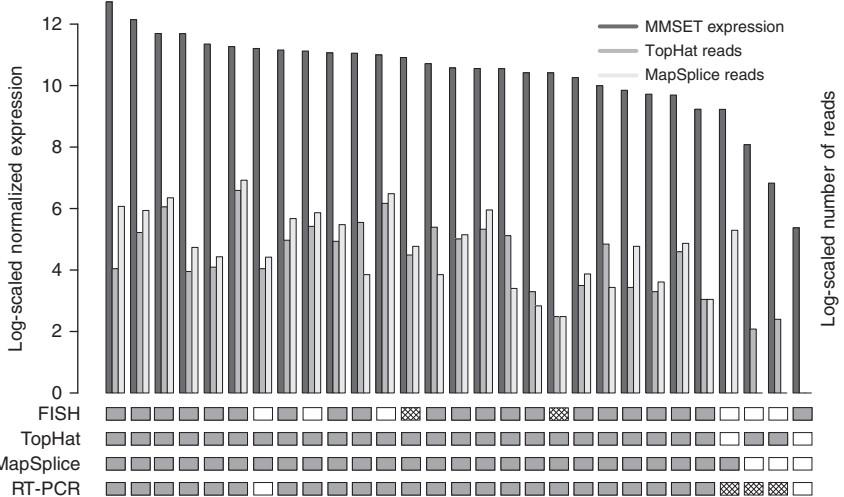

**Fig. 2** *MMSET* expression and number of supporting reads in samples described as expressing an *IgH–MMSET* fusion transcript according to at least one algorithm or to FISH. Samples are ordered by MMSET log-expression level (dark bar). For each patient, the log-number of supporting reads of each algorithm is also displayed (TopHat fusion = middle bar, MapSplice = right light bar). Results of each approach is given below, with dark rectangles representing a positive result, white rectangles representing negative result, and hatched rectangles represent samples for which no experiment was performed

able to fuse with both the heavy and light chains. Figure 3a–c represents the fusions involving IGH, kappa, and lambda respectively; (more details are displayed in Supplementary Figs. 5–7; and Ig fusions in cell lines in Supplementary Fig. 8). Among them, *Thioredoxin Domain Containing 5* (*TXNDC5* located on chromosome 6p24) was fused in 14 patient samples. In three, fusions were with *IGH* (fusion also found in two cell lines), in eight with *IGK* (and six cell lines), and in three with *IGL*. Another new fusion involves the *Beta 2 microglobulin* gene (*B2M*, chromosome 15q21), fused with both *IGH* (six samples) and *IGK* (20 patients). The *B2M–IGH* fusion was confirmed by PCR in 14 samples (six samples found by both algorithms, and another two TopHat-specific and six MapSplice-specific samples, see Supplementary Fig. 3b). Other recurrently fused genes include *FOSB* (FBJ murine osteosarcoma viral oncogene homolog B, located on chromosome 19q13 in five patients with *IGH*, three with *IGK*), *JUND* (JunD proto-oncogene, located on chromosome 19p13, in three patients with *IGH*, 15 with *IGK*); and JUN (Jun proto-oncogene, located on chromosome 1p32, in two patients with *IGH*, 11 with *IGK* and, one with *IGL*).

Though *MMSET* is the most recurrent *IGH* partner (fusion occurring in 25 patients and 19 cell lines), its fusion with light chains was not identified. Similarly, *WWOX* (tumor suppressor gene located on chromosome 16q23) is recurrently fused with *IGH* (four patients and six cell lines) but not with *IGL* nor *IGK*. Of note, an *FGFR3–IGH* fusion was also found in two patients presenting an *IGH–MMSET* fusion, suggesting that both arms of the balanced translocation can produce fusion genes.

**Identification of novel fusions**. Overall 495 genes were involved in all fusions found by both algorithms after filtering (Supplementary Data 1). Most prevalent fusions implicated genes located on chromosome 19 with partners located on the same chromosome arm that might reflect tumoral genome instability. Also frequent are fusions implicating *IGK* and multiple partners including oncogenes and tumor suppressors. We also observe a significant correlation between frequency of fusions and number of genes per megabase in each chromosome (Fig. 4) with exception of Chromosome 2 and chromosome 14 which have more fusions than expected (and can be explained by the presence

of *IGK* and *IGH* and our previous findings), and chromosome 17 which has less than expected (and can be explained by the frequent deletion 17p in MM patients).

For the most prevalent fusions, we checked the association of presence of a fusion with copy number status based on additional SNP-array data obtained from same patient tumor samples. Except in some cases of intra-chromosomal fusions, we could not detect any association between these features, ruling out the possibility of fusions resulting from germline copy-number variants.

We performed a functional enrichment study of all genes involved in at least one fusion (495 unique genes) using GSEA from Broad Institute (see "Methods" section). Most enriched pathways included *TNFα* signaling via *NFκB* (adjusted *P*-value from GSEA ~10e-31, based on hypergeometric distribution comparing the intersection list to the number of genes in the pathway and the 495 entry genes), *P53* pathway (adjusted *P*-value ~10e-15), and apoptosis (adjusted *P*-value 10e-11). Results were similar when the analysis was restricted to Ig partners. Functional enrichment study of recurrent fused genes reported NFA signaling via *NFκB* (adjusted *P*-value ~10e-21), hypoxia (adjusted *P*-value ~10e-10), and P53 pathway (adjusted *P*-value 10e-10) as most enriched pathways. (The full list of enrichment results is available upon demand.) These pathways have already been identified as particularly important in myeloma therefore confirming that fusion transcript can play a significant role in MM and may result from a non-stochastic process or undergo selective pressure[16, 17].

**Fusions in high hyperdiploid patients**. High hyperdiploid patients (53 or more chromosomes, 51/255 patients) have on average four fusions. In total, 45% of these are intra-chromosomal fusions, amongst which 30% involve *IGH* and *IGK* with partners described above (*B2M*, *TXNDC5*, *JUN*, etc.), and 54% involve chromosome 19. In particular, *TPM4* (tropomyosin 4, located on chromosome 19p13) appears as a frequent fusion partner, both with inter-chromosomal partners (*ACTB*, chromosome 7p22, *UBC* on chromosome 12q24, *CCNG2* on chromosome 4q21, etc.) and with intra-chromosomal partners (*ISYNA1*, *OAZ1*, *CSNK1G2*). Intra-chromosomal fusions that do not occur on

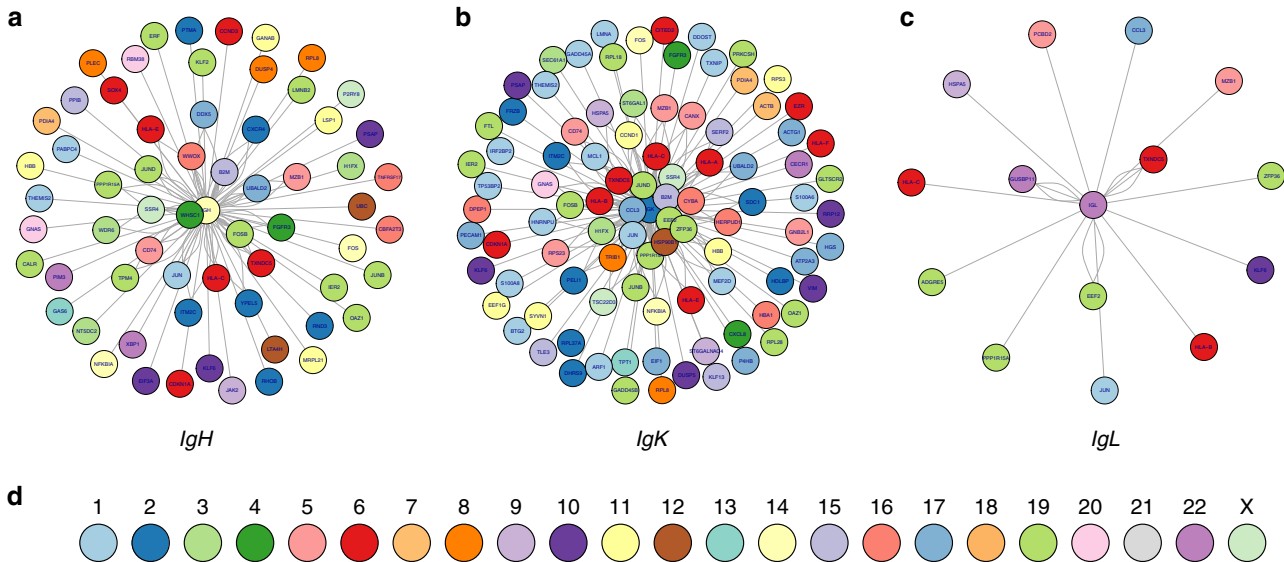

Fig. 3 a IgH fusions in patients. The center edge is IgH, each other edge represents a partner gene. There is one node per fusion. Edges closest to the center are most recurrent. Edges are colored by chromosome, as indicated in d. b IgK fusions in patients (same as in a, with center edge IgK). c IgL fusions in patients (same as in a, with center edge IgL). Supplementary Figures 5–7 show the most recurrent partners in more detail

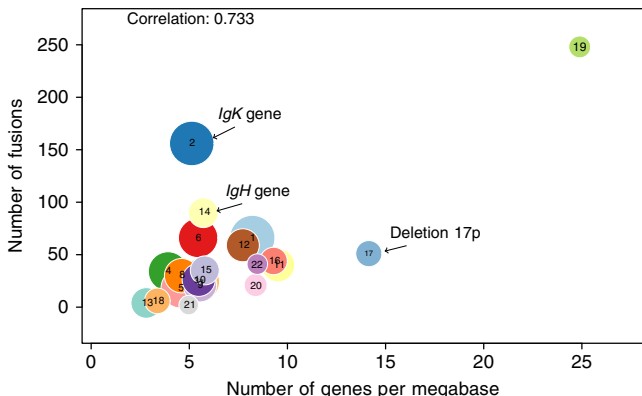

Fig. 4 Number of fusions involving genes on each chromosome compared to the chromosome's gene density. Each circle represents a chromosome, and their areas are proportional to chromosome length. Fusions involving partner genes on the same chromosome are counted twice (once for each gene). The correlation is high, with three outliers: chromosome 2 and chromosome 14 have more fusions than expected, which can be explained by the presence of the Kappa light chain on chromosome 2 and IgH on chromosome 14, and chromosome 17 has less than expected, which can be explained by the deletion 17p harbored by many patients. Once those chromosome removed, the correlation increases to 0.94

chromosome 19 (24 fusions distributed in 19 out of 51 patients) involve mostly even chromosomes (12, 14, 16, 20, and 22) with the exception of a *MAFK–ACTB* fusion (chromosome 7) occurring in two patients. This might indicate that the processes behind small amplifications and deletions (which may result in intra-chromosomal fusion creation) and whole chromosome replication (resulting in hyperdiploidy) are not directly related.

**Expression of fused genes**. Using a log-scaled normalized expression values of the most recurrently fused genes (including *MMSET*, *TXNDC5*, and *B2M*) we evaluated the expression of

fusion gene compared to the same gene without fusion, among the cohort of patients. While fused genes tend to be over-expressed (statistically significant in 22 of the 36 studied genes: adjusted *P*-value < 0.01 based on two-sided Student *t*-test corrected for multiple testing), a clear separation pattern was only found in the t(4;14) translocation (Fig. 5), for which the distribution of the expression level of *MMSET* can easily be interpreted as a mixture of high levels for all fused patients, and low levels for all non-fused patients. Of note, except in a few *IgH–MMSET* fusions, the presence of a fusion transcript did not seem to affect the respective expression of exons on each side of the predicted breakpoint, as illustrated in a few examples in Supplementary Figs. 13 and 14.

**Fusions and prognostic impact**. Among the 36 most recurrently fused genes, two are associated with a lower progression-free survival (adjusted *P*-values from log-rank test lower than 0.01; see Fig. 6a, b): *CSNK1G2* (caseine kinase 1, gamma 2 located on chromosome 19) and *CCND1* (cyclin D1, located on chromosome 11), and another two are associated with shorter overall survival: MMSET and *BCL2L11* (BCL2-like 11, apoptosis regulator, located on chromosome 2q13), as shown in Supplementary Fig. 9a, b. These four genes remained significantly important for clinical outcome when adjusted for other known high-risk prognostic variables (ISS, presence of Del 17p and presence of t(4;14), (*P*-value from Cox proportional hazard model below 0,01; see "Methods" section and Supplementary Tables 1–4).

Finally, patients with more than 16 fusions, (greater than any number observed in cell lines, corresponding to 14 patients with clinical information) have a significantly poorer prognosis than others, both in terms of progression-free and overall survival (Supplementary Fig. 10). The majority of these patients (12/14) also harbor a homozygous deletion of chromosome 13.

## Discussion

Here, we provide the first large-scale overview of the fusion transcript landscape associated with chromosomal translocations in MM using an RNA-sequencing approach. Using a large cohort of samples from newly diagnosed patients enrolled in a prospective trial we show that fusion genes are present in almost all

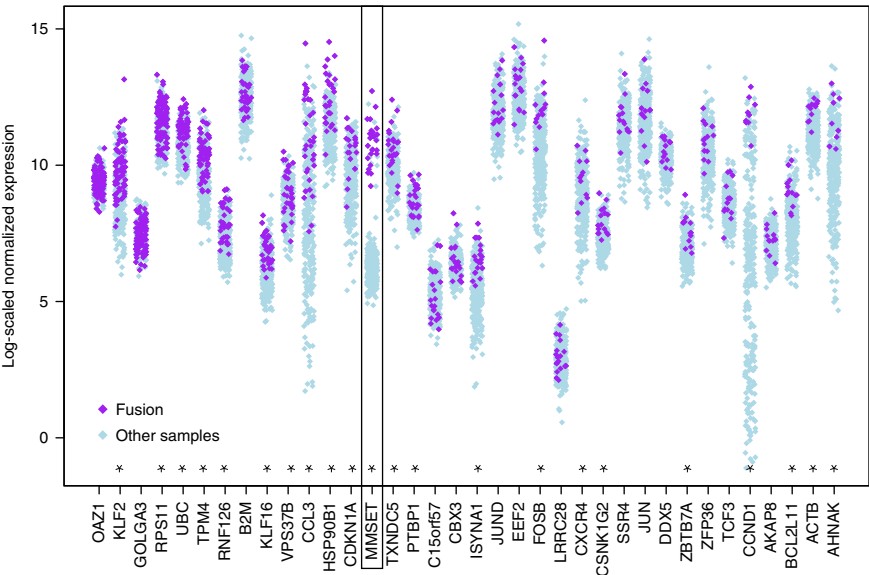

**Fig. 5** Expression of genes most frequently fused (more than 10 samples). Purple dots indicate samples for which the gene is involved in a fusion and light blue dots indicate expression value for the other samples. Generally, the expression of these genes is high in the cohort. Most genes (22 out of 36, indicated by a *star*) have significantly higher expression when fused than not (two-sided adjusted Student's *t*-test, *P*-value < 0.01), but only *MMSET* shows a very distinct separation of fused samples versus others

the patients. The well-known t(4;14) in MM provides the validation and the reliability of our method. The *MMSET* gene is affected as part of this frequent chromosomal abnormality in myeloma and forms a very important sub-group that requires both special therapeutic intervention as well as closer follow up, and provides the rationale for our initial focus on this gene. Our approach allowed identification of all except one patient carrying an *IGH–MMSET* fusion and identifying the exact break-point type, confirmed by FISH and RT-PCR. The FISH positive patient who was not detected by either algorithm could not be confirmed by PCR possibly suggesting a false positive. This confirms that RNA-seq data can identify translocations and/or intra-chromosomal rearrangements, with the possibility to precisely recognize gene recombination. Interestingly, some *IGH–MMSET* fusions were detected by the algorithms based on an unusually small number of reads, validated by PCR, but not identified by FISH, suggesting that the translocation may occur only in a subclone not detected by FISH given its lower sensitivity. This would confirm previous work reporting that t(4;14) can be present in silent subclones at diagnosis[18]. The follow-up of such patients will determine if the translocation emerges in a major clone and affects the eventual clinical outcome.

Our analysis uses a conservative approach of the intersection of two detection algorithms to limit the flaws of each method. For instance, TopHat is very sensitive to read lengths, identifying almost four times as many fusions in patients (sequenced with 50 bp reads) than in cell lines (75 bp reads). On the other hand, its annotation interface is more developed than MapSplice's and its sensitivity is higher. The differences in terms of fusion distributions between cell lines and patients should therefore be interpreted carefully, as they were sequenced with different read length. It is expected that shorter reads would result in more fusions identified, and this (as well as the greater number of patients analyzed) has to be taken into account when comparing patients' tumor cells with cell lines. However, 12% of patients presented no fusions while all cell lines presented at least one, therefore contradicting the bias one could expect from read-length. With this strategy, an average of 5.5 fusions was found in the patients (range 0–41), very similar to that observed in MM

cell lines (average 6.1). This number in MM is relatively small compared to other cancer types, for instance 49 fusions in ependymal tumors (range 34–74)[19] and 20 fusions per patient in colon carcinoma[20] but significantly higher than in pediatric B-cell precursor acute lymphoblastic leukemia (average 1 in the non-high hyperdiploid patients, the later harbor none[21]). However, as in pediatric ALL, HHMM, and hypodiploid patients exhibit less fusions than others. This suggests that MM oncogenesis pattern might be similar but extended over the longer life-time of MM patients. This claim is supported by the observation that the number of fusions increases with the age of the patients (Supplementary Fig. 2b). Of note, these numbers do not reflect the total number of translocations present in the tumor PCs, since many of them do not produce fusion transcripts. However, it is interesting to note that patients with the highest number of fusions did present a worse outcome (Supplementary Fig. 10), a finding correlating with reported impact of the mutation load and the chromosomal instability with survival. The fact that cell lines display more fusions on average with less extreme cases is unexpected and might be related to the cell lines cytogenetic which only includes non-hyperdiploid and Ig-translocated MM, underlying once more the heterogeneity in MM subtypes.

High frequency involvement of Chromosome 19 in fusions, especially intra-chromosomal, is intriguing. This may be explained by a number of unique features of Chromosome 19. It has the highest gene density of all human chromosomes, more than double the genome-wide average[22] and we have highlighted the correlation between number of fusions on a chromosome and its gene density. It also has the highest density of repetitive sequences: nearly 55% of this chromosome consists of repetitive elements (the genome average is 44.8%). This difference is due mainly to an unusually high content of short interspersed nuclear elements. Specifically, *Alu* repeats make up 25.8% of the chromosome, compared with 13.8, 13.3, 9.5, and 16.8% on chromosomes 7, 14, 21, and 22, respectively. In addition, the G+C content of the chromosome 19 is unusually high, with an average of 48% (the genome average is 41%). These repetitive sequences and high G+C content could also be responsible for a greater number of intra-chromosomal fusions.

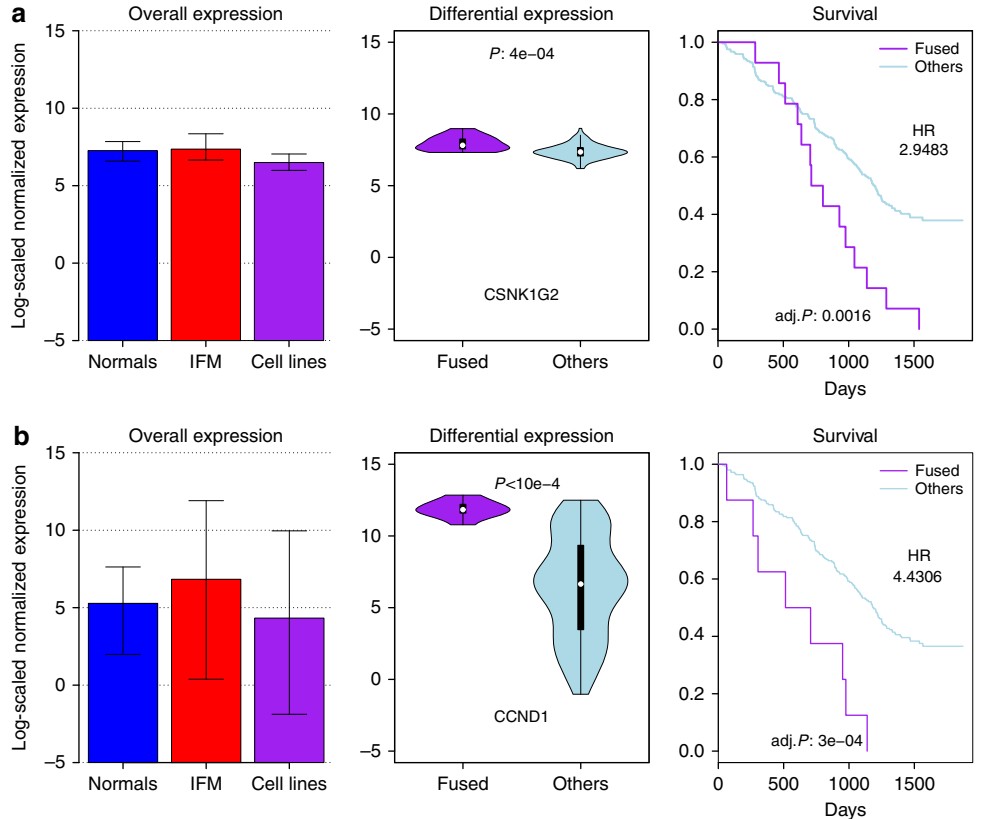

**Fig. 6** Two genes with clinical impact. *CSNK1G2* **a** and *CCND1* **b**. In each row, the left panel shows the expression of the gene in the normal samples, in the IFM cohort and in the cell lines, error bars indicating the 90% distribution intervals. The central panel shows the distribution of gene expression within the IFM cohort, comparing patients with fusion (purple, (**a**) n = 14, (**b**) n = 8), and patients without fusion (blue). The right panel gives the Kaplan–Meier estimates of the event-free survival in the two previous subgroups, as well as the log-rank *P*-value adjusted for multiple testing on the 36 most frequently fused genes, and the hazard-ratio estimates from the Cox-proportional hazard model fitted with three high-risk known variables: ISS, presence of deletion 17p, and presence of t(4;14) translocation

Regarding the nature of the fusions, 29% of them are recurrent: observed in at least two patients and up to 115 patients. This finding supports either a non-stochastic generation mechanism, or the existence of a selective pressure for fusions in the tumor clones. In total, 36% of fusions involve one of the three Ig genes (30% *IGH*, 66% *IGK*, 4% *IGL*). This incidence highlights the role of the Ig gene rearrangement processes (somatic hypermutations, class switch recombination, receptor editing) in the generation of these translocations. The fact that *IGK* is more frequently involved than the *IGL* gene (48% of kappa phenotype express a kappa-fusion, while only 14.5% of lambda phenotype express a lambda fusion) reflects probably the physiological first rearrangement of kappa. This raises the question of the translocation timing—either early during the B-cell maturation, or later through a receptor re-editing. Our findings in terms of frequent fusions involving the light chains also highlights the need to further investigate 2p11 and 22q11 loci using FISH for recurrent translocations and for possible oncogene activation through translocations. It is important to note that *IgH* and *IgK* related fusions reflect the biology of the plasma cells as well as the disease process. This is further supported by the observation of significantly increased involvement of *IgK* compared to *IgL*. Our observation that high hyperdiploid patients have significantly less *Ig* fusions than others confirms previous report analyzing translocations using FISH[23], and shows that all *Ig* fusions, including those involving light chains, in particular kappa, are affected. Hence, the current widely accepted model of myeloma oncogenesis describing hyperdiploidy and *IGH* translocations as the

two different pathways is supported, but also modified by the fact that all Ig translocations, not only *IGH*, are involved.

Among the Ig translocations, and besides the well-characterized *IGH–MMSET* fusion, a novel recurrent partner gene, *B2M* was identified, fusing with either *IGH* and *IGK* and which was confirmed by RT-PCR. *B2M* protein concentration in the blood has been used as a prognostic marker for many years[24, 25] but this gene has never been previously identified as a potential partner in fusion genes. However, *B2M* fusions were not clearly associated with over-expression of the related protein. Another interesting recurrent Ig fusion partner is *TXNDC5*. Studies have shown that this gene might be implicated in a wide range of solid tumors, acting as a tumor-enhancing gene since its over-expression is associated with an increase in proliferation and migration[26]. The role of *TXNDC5* in MM biology remains to be investigated, even if fusions involving this gene do not result in significant over-expression of the protein when compared to absence of fusions. However, *TXNDC5* is clearly over-expressed in myeloma samples when compared to normal plasma cells (Student two-sided *t*-test *P*-value < 10 e−5, see Supplementary Fig. 11). This apparent discrepancy between fusions and over-expression highlights the complexity and heterogeneity of mechanisms driven by translocations.

*TPM4* was involved in 92 fusions (6.5% of the total amount of fusions) with 32 different partners (most frequent *ISYNA1* on chr19p13, suggesting a read-through fusion, and *UBC* on chr12q24, 17 occurrences each) in 59 patients. This gene has been identified as a recurrent fusion partner in unusual childhood

extramedullary hematologic malignancy with natural killer cell properties[27]. When fused with a partner on chromosome 19, this gene is in fact associated with a better overall prognosis (log-rank test $P$-value 0.02).

*MYC* locus 8q24 is recurrently rearranged in MM (up to 50% of MM cases) with different partners, *IgH* being the main recurrent one. Of note, in these reports, *MYC* rearrangements appear to occur in the non-coding genome and to involve mainly a super-enhancer or enhancer of the second partner (*IgH*, or other partners) leading to *MYC* over-expression[28, 29]. In that setting, RNA sequencing would only identify high level of *MYC* transcript but not the fusion per se. Therefore, it is not surprising that fusion events involving *MYC* were not identified by us at the RNA level. Finally, we also identified recurrent fusions impacting clinical outcome, in addition to the well-known *IGH–MMSET*, in particular those involving *CCND1*, *CSNK1G2*, and *BCL2L11*, all these genes having been described in tumors biology[30, 31].

In conclusion, our study provides the first overview of fusion transcripts in MM and indicates a complex and heterogeneous fusion gene landscape supported by non-stochastic mechanisms. We show that in addition to the immunoglobulin genes that remain central, the landscape of fusion transcripts is much wider, and includes novel recurrent fusions, involving fundamental pathways and impacting clinical outcome.

## Methods

**The IFM-DFCI cohort**. 383 patient samples from the IFM-DFCI 2009 cohort[32], included from November 2010 to November 2012 (https://clinicaltrials.gov) were initially sequenced in three batches following the protocol described below. All the samples were collected at diagnosis in France and processed at the University Hospital of Nantes. All included patients were newly diagnosed with symptomatic MM based on International Myeloma Working Group 2003 Diagnostic Criteria[33]. All patients signed an informed consent form approved by the Toulouse Ethics Committee. Patients were younger than 65 years old, with a mean age of 56.6 years (median 58) and were 57% males and 43% females. ISS stages were 31% stage I, 47% stage II, and 22% stage III: with a median beta 2 microglobulin level of 3.5 mg/l. Finally, patients could also be divided into 63% isotype IgG, 21% isotype IgA and 16% with no heavy chain, and in 68% Kappa phenotype against 32% Lambda. In this study, only the first two sequencing batches were analyzed, with eight samples excluded because of their low number of reads (below 5 million reads), and another five were removed following a segfault error in MapSplice algorithm. In the end, 255 patient samples were analyzed for fusion detection.

**RNA preparation and sequencing**. Patients selected for sequencing had tumor purity above 90%. After extraction, RNA quantity was evaluated using the Qubit RNA Assay Kit (Life Technologies, Carlsbad, CA) and RNA quality was determined on the Bioanalyzer using the RNA Pico Kit (Agilent, Santa Clara, CA). We used 100 ng of total RNA for each sample. Next, library preparation was done with NEBNext Ultra RNA Library Prep Kit for Illumina (New England BioLabs, Ipswich, MA), was converted into a DNA library following the manufacturer's protocol, with no modifications. Library quantity was determined using the Qubit High Sensitivity DNA Kit and library size was determined using the Bioanalyzer High Sensitivity Chip Kit (Agilent). Finally, libraries were put through quantitative PCR using the Universal Library Quantification Kit for Illumina (Kapa Biosystems, Wilmington, MA) and run on the 7900HT Fast quantitative PCR machine (ABI, Grand Island, NY). Libraries passing QC were diluted to 2 nM using sterile water, and then sequenced on the HiSeq 2000 system (Illumina, San Diego, CA) at a final concentration of 12 pM, following all manufacturer's protocols. The patients retained in this study had a sequencing depth between 9.4 and 150 million reads. As a sanity check, we verified that the number of fusions identified in each patient was not correlated with the sequencing depth (Supplementary Fig. 12).

**FISH experiments**. Sorted plasma cells were fixed in Carnoy's fixative and stored at −20 °C until hybridization. After slide preparation, they were denatured in 70% formamide for 5 min, dehydrated in 70, 85, and 100% ethanol series. The probes specific for the t(4;14), t(11;14), 17p, and t(14;16) were purchased from Abbott Molecular and denatured separately for 5 min at 75 °C. After denaturation, the probes were dropped on the plasma cells and hybridized overnight at 37 °C. Then, coverslips were removed and the slides were washed for 2 min in 2× SSC-0.1% Triton at 75 °C.

**Real-time quantitative PCR confirmation**. To confirm the presence of fusions, we used extracted RNA from purified CD138 positive myeloma cells from identified patients' samples and from the t(4;14) MM cell lines KMS11, NCIH929, and OPM2 harboring the three different breakpoints MB4-1, MB4-2, and MB4-3 respectively[34]. Patients were selected for validation if they were classified as t(4;14) by at least one approach (FISH, TopHat or MapSplice) and if sufficient amount of RNA was available for RT-PCR (26 patients in total).

For the cell lines, RNA was extracted by using the RNeasy plus mini kit (Qiagen, Austin, Texas, USA) and cDNA was obtained with High Capacity cDNA Reverse Transcription Kit from Applied Biosystem (Foster City, California, 94404, USA). Quantitative PCR was performed with TaqMan Gene Expression Master Mix (Applied Biosystem, Foster City, California 94404, USA) on a 7300 Real-Time PCR System (Applied Biosystem Foster City, California, USA).

For all experiments we used the housekeeping gene GUS expression as a control (5′-CCGAGTG AAGATCCCCTTTTTA-3′ and 5′-GAAAATATGTGGTTGGAG AGCTCATT-3′). To confirm the three *MMSET* breakpoints we used targeted primers designed by Chandesris et al.[35],: 5′-TTG CAA GGC TCG CAG TGA C-3′ (IMU), and 5′-ACC ACG GTC ACC GTC TCC TCA-3′ (JH), combined with 5-TCT GAA CAG AAA GGG GAC TCT GC-3′ and 5′-AGA ACG GAA GCA TCT GGG CTG GAT-Tamra (MB4-1), 5′-TTC AAC AGG TGG TCT TTG TCT CTT C-3′ and 5′-TCC AGC TAA GAA AGA GTC TTG TCC AAA CAC T-Tamra (MB4-2), and 5′-GGG GCG TCA CCA AAG AAC TG-3′ and 5′-CAG AAA AAG AGT GCA CGC CAG TAT CAC G-Tamra (MB4-3). For the *IgH*-Beta2microglobulin fusion, we used the HUMAN GENOME Project reference (Ensembl version hg19) to design the following primers: *B2M* 5′TGC TCG CGC TAC TCT CTC TTT3′ and *IGHA* 5′GAG GCT CAG CGG GAA GAC CTT G3′.

**Fusion detection**. To detect novel fusions, we used two fusion-detection algorithms on each sample separately: TopHat (v2.0.14), and MapSplice (v2.1.9). While there is no consensus at present about the best computational approach for identifying fusions from RNA-Seq data, these two algorithms are among the most widely adopted. We made an effort to set the user-specified parameters as similarly as possible. We aligned reads by Bowtie 1[36] using the human Ensembl GrCh38 reference genome and its associated annotation. We turned on fusion-search options and required a minimum of three reads supporting a fusion candidate in order to validate it. We set the anchor length to 13 bp, and the minimum distance between two partner genes to 100,000 bp. Additionally, TopHat required that we provide expected inner distance between reads. We chose 220 based on empirical values obtained by prior alignment using Star[37] (v2.4.0). All other parameters were left to default values.

We considered a fusion to be found by both algorithms as long as the same partner genes were involved in the same orientation, even in presence of differences in the exact breakpoint predictions. For instance, our definition considers *IGH–MMSET* and *MMSET–IGH* to be two different fusions. In this way we identified an initial set of 2501 common candidate fusions. These were further filtered as described next. The pipeline developed for this analysis is available in Supplementary Data 2.

**Fusion filtering**. We used two criteria: (a) we removed fusions which did not have at least three pairs of reads for which one end covers the junction. We implemented this through a script based on the number of reads reported by both algorithms, namely "Number of spanning mate pairs where one end spans a fusion" for TopHat, and the sum of "multiple_paired_read_count" and "unique_paired_read_count", i.e., the number of multiple and unique mapped reads mapped to the fusion and that are paired with their mates for MapSplice; (b) we removed fusions for which both partners belong to the same gene family, as we are concerned about the potentially high sequence similarity. We implemented this through a script based on the official HGNC gene-family dataset, downloaded from http://www.genenames.org/cgi-bin/genefamilies/download-all/tsv on 08/10/15. The pipeline developed for this filtering is available in Supplementary Data 2.

**Sequence homology evaluation**. For each candidate fusion (including those filtered out according to the previous criteria), we computed sequence homology scores based on TopHat's prediction of breakpoint localization. To this effect, we retrieved the 20 bp sequences before (3′) and after (5′) the breakpoint of each partner gene, and computed both 3′ (respectively 5′) sequences' homology using FASTA[38] with default parameters. Our final score, given in the last column of Supplementary Data 1, is the minimum of both 3′ and 5′ E-score as obtained by FASTA. Those E-scores can be interpreted as a P-value for sequences to not share similarities: the smaller the E-score, the higher the homology. The code used for this computation is available in Supplementary Data 2.

**Geneset enrichment**. We performed gene-annotation enrichment analysis using The Gene Set Enrichment Analysis tool from the Broad Institute (v5.1)[39]. All fused genes, unique partner genes from *Ig* fusions and recurrently fused genes (involved in at least two fusions) were used (list provided as Official Gene Symbol identifier under the Gene Identifiers List in GSEA. Annotation was restricted to Homo tissue compendium (Novartis), and summary results limited to Hallmark gene sets.

**Fusion distribution**. We compared the distribution of the number of fusions in the cell lines and in the IFM-DFCI cohort via a permutation test of the Kolmogorov–Smirnov statistic by generating 99999 permutations under the null hypothesis of no difference.

We compared the number of fusions per individual across patient subsets of interest. We evaluated significance of differences using permutation tests where, for each comparison, we generated 9999 permutations under the null hypothesis of no difference. In Fig. 1b, patients with at least one Ig fusion were compared to patients with no Ig fusion conditional on their having at least one fusion. In Fig. 1c, patients with at least one translocation detected by FISH were compared to other patients, and in Fig. 1d, high hyper-diploid patients (HMM) were defined as having at least 53 chromosomes and were compared to patients with less than 52 chromosomes.

**Gene expression**. We evaluated gene expression using the same RNA-Seq data used to detect fusions. We normalized the data using the standard procedure from the *limma* R package[40]. We displayed log-scaled normalized expression values to compare the partner gene expression of samples with fusions to other samples. Because we only had relatively small numbers of fused samples, we limited these tests to the 36 most recurrent partner genes. We evaluated differential expression using two-sided Student's *t*-tests based on normalized expression values.

**Survival analysis**. For a subset of 232 patients from our cohort we have collected relapse, death and cytogenetic events including t(4;14), t(11;14), and t(14;16) translocations. This information was last updated in April 2016, with the release of the phase-3 clinical trial results, over 7 years after the first patients' diagnosis. We computed Kaplan–Meier curves and *P*-values from log-rank test using the *survival* package in R[37] (version 2.38). For the four genes that we identified as significantly impacting survival, we further proposed a multivariate analysis including three of the most relevant known high-risk prognosis variables: ISS, presence of deletion 17p, and presence of t(4;14) translocation. 16 patients are excluded for this analysis for missingness of at least one of these variables. Cox proportional hazard models were adjusted using the *survival* package in R (Version 2.41-3; https://github.com/therneau/survival), and effects tested using anova from the *limma* package.

**Data availability**. The authors declare that all data supporting the findings of this study are available within the article and its supplementary information files. Data can be accessed from the dataverse portal with doi:10.7910/DVN/RFL6TK (https://dataverse.harvard.edu/dataset.xhtml?persistentId=doi:10.7910/DVN/RFL6TK). Additional information are available from the corresponding author upon reasonable request.

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

## Acknowledgements

This work was supported by NIH grants PO1-155258 and P50-100707 (NM, H.A.-L., KA and G.P.); Department of Veterans Affairs Merit Review Award 1 I01BX001584-01 (NM) and the CAPTOR program. The CRCT Team 13 is labeled by ARC. G.P. was supported by NC1 P30-006516-51. We thank the Intergroupe Francophone du Myelome for providing patient samples and clinical data, and Dr. Keats J.J. for providing RNA-Seq data from MM cell lines.

## Author contributions

N.M. and H.A.-L. conceived the project. A.C., R.S., M.K.S., S.R.d.P., L.B., G.P., J.C., N.M., H.A.-L. analyzed the data. R.S. performed the PCR experiments. E.B., M.L.C., S.M., P.M., M.A., and H.A.-L. provided samples and clinical data. A.C., R.S., J.C., N.M., and H.A.-L. wrote the manuscript which was reviewed and edited by the other co-authors.

## Additional information

**Competing interests:** The authors declare no competing financial interests.

