## [Peer Review File · Nature Communications]

Reviewers' Comments:

Reviewer #1 (Remarks to the Author)

Multiple myeloma is a cancer of the plasma cells and frequently carries translocations involving immunoglobulin loci, leading to activating transcript fusions. Cleyen et al took to RNAsequencing of a large cohort of multiple myeloma samples to perform an unbiased characterization of the fusion gene landscape. The size of the cohort is tremendous, including 255 patient samples and 71 cell lines. The paper currently boxes below its weight in terms of impact. It is unclear what the advance is that is being reported or how to interpret the most exciting findings. In addition, the analysis could be improved. In this reviewer's opinion, the raw dataset needs to be made publicly accessible in order to warrant publication.

Major comments

1. The paper is currently written for a multiple myeloma audience which limits the interest for the broader audience of Nature Communications. The nuances of the immunoglobulin genes, such the different variants (kappa, lambda, etcetera), and other factors relevant to fusion transcripts in multiple myeloma need to be much better introduced. FGFR3 is a very important oncogene which is frequently activated through the t(4;14) but this is not clear from the current writing. The Results reads as a summary of the analysis and lacks interpretation. A thorough rewrite is needed to improve the readability and excitement over this manuscript. For example, why is it relevant to report the number of fusions per cytogenic group? How do the findings relate to the genomic characteristics of these groups?
2. Details of the sequencing dataset are needed to be able to interpret the findings, for example sequencing yield per library, tumor purity, QC parameters etc. The numbers reported, such as the number of fusions per sample, are affected by both coverage and purity and normalization is needed to adequately compare between samples. Read length has a considerable impact on the sensitivity and specificity of fusion transcript detection and the difference in read length between cell line and patient samples sequencing needs to be considered in the interpretation of the results. Where there are any differences in library construction such as median fragment size?
3. The analysis needs to be reproducible. Additional details on how the analysis was performed are required to improve reproducibility. For example, what does the gene family script do?
4. Gene homology may result in incorrect gene fusion predictions. The gene family script may partially address this but provides an indirect approach to this potential confounder. Gene homology should be directly measured and used to filter the list.
5. Fusions may result from somatic cancer-specific events but can also result from germline copy number variants. This possibility needs to be considered at least for the most frequent events.
6. Additional means to identify the most cancer-relevant fusions include comparison of the number of reads supported fusion transcripts versus the number of reads supporting the wild type transcripts (for example # IGH-MYC reads versus # IGH or # MYC reads). Alternatively, exon expression levels on either side of each breakpoint could be compared as activating fusions typically result in overexpression of the exons that are part of the fusion.
7. Supp Table 1 lacks sample IDs limiting the interpretation of the fusion list. The Table reports 905 fusions while the manuscript reports 1408 + 436 fusions. It is unclear what additional filtering was performed to achieve this Supplementary Table.
8. MMSET is introduced as the most frequent fusion partner gene. This gene needs further introduction. Why is specifically this gene highlighted and not other seemingly interesting/relevant genes such as CCND1, MYC, etc?
9. The analysis of prognostic impact needs to be corrected for multiple testing. Multivariate analysis is needed to show whether the significance of the two gene fusions reported to have prognostic impact are independent from other known prognostic variables.

Reviewer #2 (Remarks to the Author)

Cleynen et al. characterized the fusion gene landscape of multiple myeloma by RNAseq. The study generated a huge amount of data and, while interesting; there are some issues that need to be addressed.

Data in Supplementary figure 1 showed a significantly discrepancy between algorithms, with only a small fraction of overlapping results. These differences seems beyond expected. Can the authors elaborate on this?

Previous studies performed by Dr. Gareth Morgan's group have shown recurrent genomics abnormalities involving MYC with several partners, including IgH. In this study, MYC is not listed as one of the major findings at the RNA level. Do the authors have any plausible explanation about this apparent discrepancy between studies?

The authors proposed tumor genome instability recurrently involving chromosome 19. Chromosome 19 is not particularly instable at the copy-number analysis. These findings should be further tested and some results validated in order to rule out technical or bioinformatics artifacts.

The authors showed that cases with fusions in CSNK1G2, JUND, AKAP8 and BCL2L11 were associated with either poor outcome or lower PFS. However, as the authors stated, the fusions don't seem to affect the expression of those genes. Any possible explanation of this apparent discrepancy?

Citations are formatted differently across the manuscript.

Reviewer #1

General comments

1. **The size of the cohort is tremendous, including 255 patient samples and 71 cell lines. The paper currently boxes below its weight in terms of impact. It is unclear what the advance is that is being reported or how to interpret the most exciting findings.**

Important results includes 1. Fusion transcripts are frequent (>5 per patient) in MM; 2. Besides the well described IgH gene, kappa or lambda light chains are frequently involved as the main partners of fusions (a third of the overall fusion gene load); 3. We report number of novel recurrent fusion genes; and 4. Fusion genes have significant biological and clinical impact (both progression-free and overall survival). We have now highlighted these in abstract and discussion.

2. **In this reviewer's opinion, the raw dataset needs to be made publicly accessible in order to warrant publication.**

We totally agree and will make all the raw datasets publicly accessible very shortly.

Major comments

1. **The paper is currently written for a multiple myeloma audience which limits the interest for the broader audience of Nature Communications. The nuances of the immunoglobulin genes, such the different variants (kappa, lambda, etcetera), and other factors relevant to fusion transcripts in multiple myeloma need to be much better introduced. FGFR3 is a very important oncogene which is frequently activated through the t(4;14) but this is not clear from the current writing. The Results reads as a summary of the analysis and lacks interpretation. A thorough rewrite is needed to improve the readability and excitement over this manuscript.**

We have now entirely revised the manuscript to convey the information directed at a broader audience and to explain the significance and nuances associated with myeloma in the context of the results presented here.

2. **Details of the sequencing dataset are needed to be able to interpret the findings, for example sequencing yield per library, tumor purity, QC parameters etc. The numbers reported, such as the number of fusions per sample, are affected by both coverage and purity and normalization is needed to adequately compare between samples. Read length has a considerable impact on the sensitivity and specificity of fusion transcript detection and the difference in read length between cell line and patient samples sequencing needs to be considered in the interpretation of the results. Where there any differences in library construction such as median fragment size?**

We now add a more detailed discussion on the influence of sequencing framework on our results. We show for instance that the number of fusions identified is not influenced by the sequencing yield (the correlation is of the order of 0.3, and we have added a supplementary figure 11 that illustrates our point). Moreover, the tumor purity was above 90% in all patients and hence the variation between samples should not explain the difference in number of fusions identified.

On the other hand, read length has indeed a considerable impact on the number of fusions identified by the algorithms, especially in regards to TopHat. Since all patients had the same sequencing read-length, we now insist more on this difference when comparing cell-lines and patient fusion landscape.

3. **The analysis needs to be reproducible. Additional details on how the analysis was performed are required to improve reproducibility. For example, what does the gene family script do?**

We have now added our analysis pipeline scripts as a supplementary material 2 for the sake of reproducibility. The gene family script compares the families to which each of the fusion partner genes belong. If those families (defined by the official HGNC gene family dataset downloaded on 08/10/15) are identical, the fusion candidate is discarded for further analysis. Of note, in the supplementary material presenting our complete list of fusions identified, all fusions are included, the fifth column indicates how many of those candidates were discarded.

4. Gene homology may result in incorrect gene fusion predictions. The gene family script may partially address this but provides an indirect approach to this potential confounder. Gene homology should be directly measured and used to filter the list.

Our supplementary table 1 now includes a measurement of the homology of the sequences that surround the breakpoint. More precisely, for each of the 905 unique fusions, using Tophat-fusion's results, we have extracted the 20 bp before (respectively after) the first gene's breakpoint as well as the 20 bp after (respectively before) the partner gene's breakpoint and compared those sequences using the FASTA software with default parameters. The resulting E-score can be interpreted as a p-value for sequences to not share similarities; the smaller the E-score, the higher the homology. The scripts used to obtain such scores are included in the supplementary material presented in the response to comment number 3 above.

We have however decided not to filter out candidates based on this score as we aim at describing the broad fusion landscape in our cohort. Moreover very few fusion candidates have a significantly low E-score. Having identified them, we leave it to the readers' discretion to define their own threshold.

5. Fusions may result from somatic cancer-specific events but can also result from germline copy number variants. This possibility needs to be considered at least for the most frequent events.

For some of the most prevalent fusion candidates (for instance GOLGA3-RSP11, OAZ1-KLF2 or C15orf57-CBX3) we checked the copy-number status of the tumor cells with SNP-arrays. We did not identify any correlation between copy-number variants and presence of a fusion. As an example, GOLGA3 is deleted in 9 samples out of 88, its partner RSP1 has amplifications in 47 samples, other fused patients show no CNV. While this does not mean that those fusions are driver events in MM, we believe that in this instance the likelihood of those events resulting from germline copy-number variants is very small.

6. Additional means to identify the most cancer-relevant fusions include comparison of the number of reads supported fusion transcripts versus the number of reads supporting the wild type transcripts (for example # IGH-MYC reads versus # IGH or # MYC reads). Alternatively, exon expression levels on either side of each breakpoint could be compared as activating fusions typically result in overexpression of the exons that are part of the fusion.

We have now analyzed difference in coverage between exons on both sides of the fusion for CSNK1G2, AKAP8, BCL2L11, CCND1 and WHSC1 (genes having prognostic impact). Except in a few cases for WHSC1, we have not observed any clear difference in coverage between exons (on both sides of the fusion) indicating that fusion transcripts are not over-expressed compared to the wild type transcripts (examples of these results are presented in new supplementary figure 12 and 13). Of note, these kind of results should in general be difficult to obtain for various reasons: some genes are affected by multiple fusions at different exon locations, some fusions affect the first exon of a gene therefore expression on both side of the breakpoint cannot be compared, RNA-sequencing is not without bias and the coverage is never uniform, RNA degradation leads to lower coverage on the last exons, etc.

7. Supp Table 1 lacks sample IDs limiting the interpretation of the fusion list. The Table reports 905 fusions while the manuscript reports 1408 + 436 fusions. It is unclear what additional filtering was performed to achieve this Supplementary Table.

Supplementary table 1 presents all fusion candidates (905); column D indicating how many patients were first identified to carry this fusion by both algorithm, while column E indicating the remaining number of patients with that specific fusion after filtering.

For instance, the IGL-IGLL5 candidate fusion (line 6) was identified in 37 patients (column D), but none were included for further analysis (column E) as IGL and IGLL5 belong to the same gene family. Moreover, as stated in comment number 4 above, we have now added a column F containing sequence homology scores

So overall there are a total of 1408 (sum of column E) fusions in the patients of which 905 are unique candidate fusions. 436 fusions represent total numbers of fusion identified in cell lines.

8. MMSET is introduced as the most frequent fusion partner gene. This gene needs further introduction. Why is specifically this gene highlighted and not other seemingly interesting/relevant genes such as CCND1, MYC, etc?

MMSET gene is affected as part of one of the very common chromosomal abnormalities, t(4;14), in myeloma. This translocation is associated with poor survival and forms a very important sub-group that requires both special therapeutic intervention as well as closer follow up. That is why it is of particular interest in the fusion landscape of multiple myeloma. Moreover, as patients are routinely screened for the presence of a t(4;14) translocation, the IGH-MMSET fusion represents a «gold standard» to validate our fusion detection approach. This is now described and discussed clearly and in a way as mentioned in the response #1 above.

9. The analysis of prognostic impact needs to be corrected for multiple testing. Multivariate analysis is needed to show whether the significance of the two gene fusions reported to have prognostic impact are independent from other known prognostic variables.

We have now modified our analysis to correct for multiple testing (we removed two previously identified genes impacting overall survival), and for each significant gene we have proposed a Cox proportional hazard model to include the effect of known high-risk prognostic variables: ISS, deletion 17p and t(4;14) translocations. This analysis is presented in the manuscript and results are given in supplementary tables 1 to 4.

Reviewer #2

1. Data in Supplementary figure 1 showed a significant discrepancy between algorithms, with only a small fraction of overlapping results. These differences seems beyond expected. Can the authors elaborate on this?

We agree that the discrepancy is high. In general, the frequency of recurrent fusion transcripts is much lower than other somatic mutations. At present, 30 different methods for identifying gene fusions have been reported of which Tophat-Fusion is reported as one of the most widely used packages [PMID: 27105842 -- Nucleic Acids Res. 2016 Jun 2;44(10):4487-503. doi: 10.1093/nar/gkw282. Epub 2016 Apr 21.] and MapSplices has been shown with high sensitivity on several papers [PMID: 26862001 - Sci Rep. 2016 Feb 10;6:21597. doi: 10.1038/srep21597.].

Due to critical algorithm differences between two tools [Tophat-Fusion: 1) creating partial exons from the alignment, generated by mapping of reads to exons, 2) generation of pseudo-genes, while unmapped reads are split into shorter elements, and mapped on the genome, 3) detection of chimeras, if reads fragments map in a steady way with fusions, and 4) filtering to eliminate chimeras associated with multi-copy genes, or repetitive sequences and MapSplice: it splits each

read into a set of consecutive elements, and then exon alignment is performed. By using the knowledge of other aligned elements, it aligns the elements, which are not aligned in the previous step. Second, it uses two statistical measures to check the quality of the splice junctions identified in the first step. These two measures are: 1) “anchor significance”, produced by an alignment of maximum significance, resulting from long anchors on the both sides of splice junctions, and 2) “entropy”, which is calculated by the multiplicity of splice junction locations (PMID: 26862001)] and the known significant heterogeneity in MM, a higher level of differences may be expected in primary human samples with huge sample size. Moreover, recent studies showed that meta-analysis/ensemble based approaches improves the sensitivity/specificity. Therefore, we combined two methods to catch the real true positives which may cost us to miss some true positives but obviously help us to remove many false positives from the analysis. We have now added a brief summary of above explanation in our discussion.

2. Previous studies performed by Dr. Gareth Morgan’s group have shown recurrent genomic abnormalities involving MYC with several partners, including IgH. In this study, MYC is not listed as one of the major findings at the RNA level. Do the authors have any plausible explanation about this apparent discrepancy between studies?

We are aware of publications by Drs. Morgan, Kuehl and Bergsagel on MYC rearrangements in MM. These publications have reported that the locus 8q24 (which includes MYC) is recurrently rearranged in MM (up to 50% of MM cases) with different partners. IgH is the main recurrent partner but other partners have been identified, all rearrangements leading to MYC overexpression. Of note, in these reports, MYC rearrangements appear to occur in the non-coding genome and to involve mainly a super-enhancer or enhancer of the second partner (IgH, or other partners) leading to MYC overexpression. In that setting, RNA sequencing would only identify high level of MYC transcript but not the fusion per se. Therefore it is not surprising that fusion events involving MYC were not identified by us at the RNA level: except in the case of read-through fusions, the translocation needs to occur within the gene’s boundaries in order to result in a fusion transcript. A strategy of targeted DNA sequencing in samples with high MYC expression would be better to evaluate the specific partners of MYC. However, the goal of our study was to evaluate and identify the landscape of all fusion transcript.

3. The authors proposed tumor genome instability recurrently involving chromosome 19. Chromosome 19 is not particularly instable at the copy-number analysis. These findings should be further tested and some results validated in order to rule out technical or bioinformatics artifacts.

Chromosome 19 has number of unique features. Chromosome 19 has the highest gene density of all human chromosomes, more than double the genome-wide average (Nature 2004 - <http://www.nature.com/nature/journal/v428/n6982/full/nature02399.html>). Moreover, Chromosome 19 is also unusual in its density of repetitive sequences. Nearly 55% of this chromosome consists of repetitive elements (Highest amongst all chromosomes. - the genome average is 44.8%). This difference is due mainly to an unusually high content of short interspersed nuclear elements (SINEs) on chromosome 19. Specifically, *Alu* repeats make up 25.8% of the chromosome, compared with 13.8%, 13.3%, 9.5% and 16.8% on chromosomes 7, 14, 21 and 22, respectively. In addition, the G + C content of the chromosome is unusually high, with an average of 48%. This compares with 41% as reported in the whole human genome analysis. With the largest clustered gene families, corresponding high G + C content, CpG islands and the highest density of repetitive DNA sequences indicates a chromosome rich in biological and evolutionary significance. These features may predispose it to developing fusions, especially the reported intra-chromosomal fusions, possibly explaining our observation.

We have added some discussion on chromosome 19 in the manuscript and added a figure 4 which correlates the number of fusions and the gene density of each chromosome, showing that this later feature seems most relevant to explain the high number of fusions identified on chromosome 19.

- 4. The authors showed that cases with fusions in CSNK1G2, JUND, AKAP8 and BCL2L11 were associated with either poor outcome or lower PFS. However, as the authors stated, the fusions don't seem to affect the expression of those genes. Any possible explanation of this apparent discrepancy?**

We believe that this is an interesting observation as the functional impact of fusion without expression change may suggest the possibility that the fusion product by itself plays a significant role in the biology of the tumor cell affecting adverse clinical outcome. We have now mentioned this in our discussion. Of note, in the case of JUND and AKAP8, the fusions do not seem to affect the expression of the gene, however, based on Reviewer 1's suggestion, we have removed these two genes from our manuscript as they do not resist the multiple testing and multivariate analysis adjustments.

- 5. Citations are formatted differently across the manuscript.**

We have now reformatted all the citations per Nature Communication guidelines.

Reviewers' Comments:

Reviewer #1:

Remarks to the Author:

Paper much improved. One final suggestion is to further indicate the relevance of fusing to either IgH or IgK chain be explained/interpreted (section 'Landscape of Ig partner genes')?

Reviewer #2:

None

Reviewer #1 (Remarks to the Author):

Paper much improved. One final suggestion is to further indicate the relevance of fusing to either IgH or IgK chain be explained/interpreted (section 'Landscape of Ig partner genes')?

We believe that IgH and IgK related fusions reflect the biology of the plasma cells as well as the disease process. This is further supported by the observation of significantly increased involvement of IgK compared to IgL gene which is reflective of physiological first rearrangement of kappa light chain. Our data does show the fused genes to be over-expressed to some extent, but we do not observe specific differences between IgH and IgK. We have now included this in our discussion.